# Network pharmacological insight into traditional bone healing practices of Sikkim, India

Mukunda Anuj Sharma[1,2☯], Bharat Gopalrao Somkuwar[3,4☯], Parvin A. Barbhuiya[1], Bhumika Gurung[1], Madhusmita Mahapatra[1], Firdous Fatima[3], Teresa Ningthoujam[1,4], Ashika Bhattarai[1], Bikash Rai[1], Pravin Kumar[1], Bishal Tiwari[1], Sancha Kumar Subba[5], Zyankit Lepcha[6], Purna Maya Gurung[7], Nandalal Khadka[8], Ratan Bahadur Tamang[9], Balbir Khati[10], Shribhakta Chettri[11], Ran Bahadur Rai[12], Hem Lall Sharma[13], Yam Bahadur Rai[14], Norzang Lepcha[15], Theptuk Lepcha[16], Pritiman Singh Chettri[17], Monraj Limboo[18], Theweng Gyenthen[19], Tulshi Pradhan[20], Prem Gurung[21], Prem Prashad Dhakal[22], Nanaocha Sharma[1,4], Lokesh Deb(iD)[1,4]*

1 Biotechnology Research and Innovation Council - Institute of Bioresources and Sustainable Development (BRIC-IBSD) - Regional Centre, Sikkim (Department of Biotechnology, Government of India), Gangtok, Sikkim, India, 2 Department of Zoology, Sikkim University, Gangtok, Sikkim, India, 3 Biotechnology Research and Innovation Council-Institute of Bioresources and Sustainable Development - Mizoram (Department of Biotechnology, Government of India), Aizawl, India, 4 Biotechnology Research and Innovation Council - Institute of Bioresources and Sustainable Development (BRIC-IBSD) (Department of Biotechnology, Government of India), Takyelpat, Imphal, Manipur, India, 5 Traditional Healer, Pachey Village, Samsing, Pakyong, Sikkim, India, 6 Traditional Healer, Lower Radhu, Dentam, Geyzing, Sikkim, India, 7 Traditional Healer, Upper Chuba, Phongla, Namchi, Sikkim, India, 8 Traditional Healer, Tangzi, Rateypani, Namchi, Sikkim, India, 9 Traditional Healer, Dhaje Dhara, Nazi Ruchung, Namchi, Sikkim, India, 10 Traditional Healer, Lower Rangang, Yangyang, Namchi, Sikkim, India, 11 Traditional Healer, Namphok, Yangyang, Namchi, Sikkim, India, 12 Traditional Healer, Ladam Machong, Pakyong, Sikkim, India, 13 Traditional Healer, Assam Daragong, Assam Linzey, Pakyong, Sikkim, India, 14 Traditional Healer, Titiribotey, Rorathang, Pakyong, Sikkim, India, 15 Traditional Healer, Shipgyer, Mangan, Sikkim, India, 16 Traditional Healer, Upper Gor, Lower Dzongu, Mangan, Sikkim, India, 17 Traditional Healer, Upper Singhik, Mangan, Near MSSS, Mangan, Sikkim, India, 18 Traditional Healer, Mangshila, Upper Ralak, Mangan, Sikkim, India, 19 Traditional Healer, Shakathang, Lachung, Mangan, Sikkim, India, 20 Traditional Healer, Lower Timburbong, Soreng, Sikkim, India, 21 Traditional Healer, Namcheybong, Pakyong, Sikkim, India, 22 Traditional Healer, Aritar, Khamdong Singtam, Gangtok, Sikkim, India

☯ Bharat Gopalrao Somkuwar and Mukunda Anuj Sharma contributed equally
* lokeshdeb@gmail.com; lokeshdeb.ibsd@nic.in

## Abstract

### Ethnopharmacological relevance

Sikkim is a mountainous state situated in the Eastern Himalaya region of India, which constitutes an area with rich cultural diversity, having different traditional healthcare practices and rituals. Traditional formulations for treating bone fractures are prevalent in rural areas of Sikkim.

### Aim

The present study was designed to document and analyse the traditional knowledge, practices, and medicinal plants used by traditional healers of Sikkim for the treatment

**Data availability statement:** All relevant data are within the manuscript and its Supporting Information files.

**Funding:** The author(s) received no specific funding for this work.

**Competing interests:** The authors have declared that no competing interests exist.

**Abbreviations:** FoC, frequency of citation; RFC, Relative Frequency of Citation; WP, Whole Plant; ADME, Absorption, Distribution, Metabolism and Elimination; PPI, Protein-protein Interaction; IMPPAT, Indian Medicinal Plants, Phytochemistry and Therapeutics; SMILES, Simplified Molecular Input Line Entry System notations; DAVID, The Database for Annotation, Visualisation, and Integrated Discovery (DAVID); GO, Gene Ontology; OMIM, Online Mendelian Inheritance in Man; KEGG, Kyoto Encyclopedia of Genes and Genomes.

of Bone fractures. And network pharmacological perspectives on the bone-mending properties of the medicinal plants used by the traditional healers of Sikkim.

## Method

Semi-structured questionnaires, semi-structured interviews, and guided field walks were used in this explorative study for four years in all six districts of Sikkim, India. The quantity indices frequency of citation (FoC) and Relative Frequency of Citation (RFC) are used to authenticate the most important medicinal plant species. Further, to examine the intricate relationships among drugs, targets, and diseases, we conducted network pharmacological annotations using several advanced tools, including Swiss Target Prediction, STRING, STITCH, DAVID, GeneCodis, and SwissADME.

## Results

The study documented 18 distinct traditional polyherbal formulations that incorporate 32 medicinal plant species native to Sikkim and are utilised for the therapeutic management of bone fractures. Notably, several plant species identified in this investigation, particularly those exhibiting high Frequency of Citation (FoC) values, represent promising candidates for further pharmacological evaluation targeting osteoregenerative properties. Additionally, four plant species, *Urtica parviflora* Roxb., *Saurauia napaulensis* DC., *Rubus calycinus* Wall. ex D.Don, and *Schima wallichii* (DC.) Korth., employed by traditional healers in this study, warrant prioritised phytochemical investigation due to their limited scientific exploration in existing literature. The network pharmacological annotations revealed several pathways that are directly or indirectly affecting bone development, biomineralisation, calcium signalling, endochondral ossification with skeletal dysplasia, RANK signalling, RUNX2 regulation, and Vitamin D-sensitive Ca signalling.

## Conclusion

This study systematically documents traditional treatments for bone fractures in Sikkim, highlighting 32 medicinal plants with therapeutic potential. The findings of this study will provide baseline data to address an immediate need to preserve and scientifically validate (in vivo and in vitro) indigenous ethnomedicinal knowledge. Furthermore, provide valuable insights into the development of safe and effective lead compounds by considering the biological processes, molecular functions, and cellular components involved in bone mending from natural formulations.

## Introduction

Bone fractures are a major public health problem and place a significant burden on the healthcare system. The musculoskeletal injuries are attributed to the largest disability-adjusted life years (DALYs) globally [1]. The fractures of bones and

their associated infections have long been recognised in human disease history. Shreds of evidence exist for traditional bone-setting practices performed in ancient times. In 1903, Mr Edwin Smith in Egypt, provided evidence of the earliest immobilisation treatment for bone fractures and dislocation [2].

In developing nations, Western medical concepts and practices for treating orthopaedic trauma have emerged recently [3]. However, contemporary medical approaches to treating bone fractures involve immobilising fractures with plaster or a cast, supplemented by anti-inflammatory and analgesic medications [4]. Recently, emphasis has been placed on the use of folk herbal medicines to maintain bone health and accelerate bone healing.

Traditional bone-setting practices, deeply rooted in many cultures and societies worldwide, encompass a rich array of therapeutic techniques that have been passed down through generations without proper documentation. The use of folk bone setters for treating musculoskeletal injuries is prevalent in developing nations [3,5]. The practice of ancient ortho-paedic surgeons in India can be traced back thousands of years to the Harappan civilisation, which left a remarkable legacy spanning over 4,000 years [6]. The treatment of bone fractures and orthopaedic care is described as "BHAGNA CHIKITSA" in Ayurved [7].

This deep-seated heritage encompasses a wide range of therapeutic techniques, including the reduction of fractures by splinting and casting, the practice of therapeutic trepanation and the usage of various plant-based pastes and oil mas-sages [1]

The study of traditional medical practices and remedies used by Indigenous communities has gained significant atten-tion in recent years. However, Indigenous ethnomedicinal knowledge in the Himalayan region has declined substantially [8]. Therefore, the documentation and analysis of traditional knowledge are of the utmost importance, as they not only aid conservation but also contribute to the development of novel therapeutics.

Several attempts have been made to document ethnomedicinal information from the local tribal community in the region [8–10]. However, a comprehensive study of traditional formulations, ingredients, and other uses has not been conducted to date. Therefore, the purpose of this study is to document, investigate, and analyse the traditional knowledge, practices, and medicinal plants used by traditional healers in Sikkim to treat bone fractures.

Network pharmacology integrates bioinformatics, systems biology, and pharmacology to analyse the relationships between drugs, targets, and diseases. It emphasises the concept of "multi-compound, multi-target, and multi-pathway" interactions, making it particularly suitable for studying polyherbal formulations. Phytochemical selection for network phar-macological annotations was impacted by the high frequency of citations from traditional healers in ethnopharmacological surveys, enabling a hypothesis-driven bridge from traditional use to multi-target mechanisms. This methodology identifies active phytochemicals in herbal components, maps their potential targets, and elucidates the signalling pathways they influence [11–14]. The bone-healing network pharmacology annotation of 32 polyherbal compounds from ethnopharmaco-logical studies in Sikkim provides the first baseline data toward the development of safe and efficacious drugs.

## Materials and methods

### Study area

Sikkim is a relatively small state in the north-east of India, situated in the foothills of the eastern Himalaya. The area spans 7,096 km$^2$ and lies between 27° 5' N to 20° 9' N latitudes and 87° 59' E to 88° 56' E longitudes. Three international borders surround this state: Bhutan to the east, Nepal to the west, and China to the north; it shares a state boundary with West Bengal to the south. Administratively, the state is divided into six districts and 16 subdivisions. The Mangan district (North Sikkim) is the largest, covering 4,226 Km$^2$, followed by the Gyalshing district (West Sikkim), which covers 1,166 Km$^2$, the Gangtok district (East Sikkim), 952 km², the Namchi district (South Sikkim), 750 Km$^2$, the Pakyong district, 404 Km$^2$ and the Soreng district, 293 Km$^2$. The region is predominantly covered by dense forest. The study area map was prepared using ArcGIS v10.4 (Esri lnc.) based on administrative boundary information from the Survey of India open assess data (Fig 1).

 

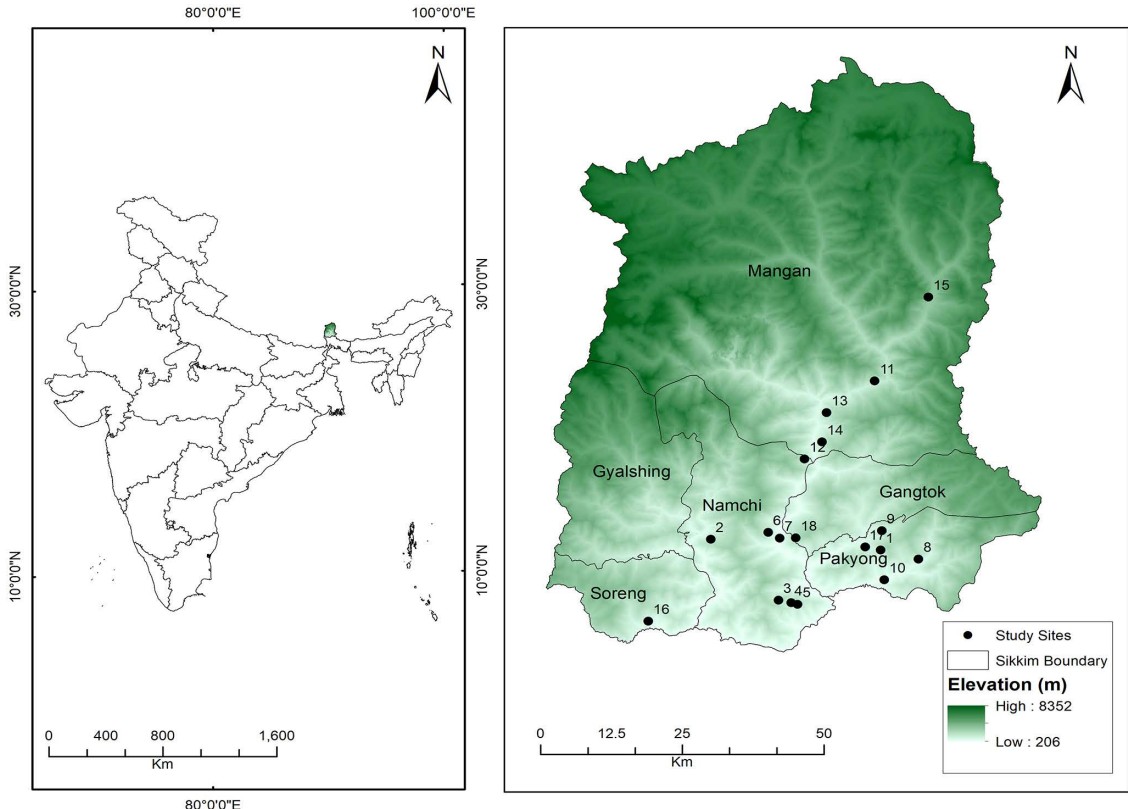

**Fig 1. Map showing study sites and demography of the traditional healers included in the study: 1. Samsing, 2. Dentam, 3. Chuba, 4. Rateypani, 5. Dhaje, 6. Yangyang, 7. Namphok, 8. Machong, 9. Lingzay, 10. Rorathang, 11. Mangan, 12. Dzongu, 13. Mangan, 14. Mangshila, 15. Lachung, 16. Timburbung, 17. Pakyong, 18. Khamdong.** Base map created using R GIS from Survey of India topographic data (2025). Base map created using ArcGIS v10.4 software.

Sikkim is a multi-ethnic State in India, with a rich and diverse culture represented by three major ethnic groups: Bhutias, Lepchas, and Nepalese. Ethnographically, the inhabitants of Sikkim exhibit affinities with Mongoloid, Aryan, and Tibeto-Burman ancestral lineages [9]. Eco-tourism, agriculture, and animal husbandry form the main pillars of the economy. Traditional healthcare practices remain integral to the primary healthcare system in Sikkim [8].

### Informants and ethno-medicinal data collection

An exploratory study was conducted to collect data, and an ethnopharmacological survey (a semi-structured, questionnaire-based, cross-sectional, descriptive study) was performed from April 2022 to May 2024 in all six districts of Sikkim (Mangan, Gyalshing, Gangtok, Namchi, Pakyong, and Soreng). Ethnomedicinal data on plants used for bone mending were collected through interviews with 26 respondent healers from different localities. The objective of the study was clearly explained in the local language (Nepali). The participants who were willing to share information provided written consent (on a consent form). All the respondent healers from all six Districts of Sikkim were considered for the study (Fig 1) with a prior research permit obtained from the Department of Forest, Environment and Wildlife Management, Government of Sikkim, Gangtok (F. No.: 78/GOS/FandED/R and E/139 dated August 3, 2022 and F. No.: 78/GOS/FEWMD/R and E/2027 dated November 7, 2023). The key informants (26 healers) were well recognised in the community for their extensive experience in providing traditional healthcare services.

All participants were interviewed in their regional language, and information was recorded regarding the specific plants they used for bone mending. The data was also collected using a pre-designed questionnaire presented in both written and audio-visual formats. The informants were asked to provide details on the parts of plants used, other non-plant materials incorporated, the mode of preparation, the methods employed (decoction, juice, infusion, or powder), the modes of administration, and the duration of treatment. Transect walks and field observation supplemented by interviews.

During the transect walks, plants were carefully observed and collected under the guidance of traditional healers. Plant voucher specimens were collected with research permits and subsequently deposited in the Herbarium for identification by the Botanical Survey of India, Himalayan Regional Centre, Gangtok, Sikkim. Data collection is done in compliance with the World Intellectual Property Organisation (WIPO) directives concerning traditional cultural expressions and in accordance with the ethical principles of the International Society of Ethnobiology (ISE) (WIPO Intellectual Property Handbook: Policy, Law and Use, 2nd ed.; WIPO Publication: Geneva, Switzerland, 2004)

## Quantitative analysis

Complete information on folk medicines for bone mending in Sikkim was collected through questionnaires and interviews. Data on each respondent and their traditional knowledge concerning bone mending- including disease condition, plant name (local, universal, scientific), formulation use, mode of application (topical and oral), the dosage form, plant part used, and duration of treatment used were compiled in a systematic scientific manner to provide comprehensive details.

The dataset obtained from the questionnaires completed during the ethnopharmacological survey was normalised in Microsoft Excel 2019. The selected standard statistical indices are relevant to the present study. The Frequency of Citation, calculated using the standard formula, is given below.

### Frequency of Citation (FoC)

Descriptive statistical calculations for FoC determine the relative importance of various plant species. The medicinal plants with the highest citation frequency may provide valuable insight into the significance of species, as determined by informant consensus, for probable new species of interest. FoC is calculated by following the formula described elsewhere [10,15].

$$FoC = \frac{\text{Number of citation of that particular species}}{\text{Total number of all citation for all species}} \times 100$$

### Relative Frequency of Citation (RFC)

The descriptive statistical calculations for RFC values offer insights into the overall significance of specific medicinal plants as reported by the informants. The RFC values are determined using the formula described elsewhere [10,15].

$$RFC = \frac{FC}{N}(0 < RFC < 1)$$

Where "FC" denotes the total number of informants citing a particular species, and "N" denotes the total number of informants who participated in the study.

### Network pharmacological annotations

**Phytoconstituents.** Phytoconstituents and metabolites of the bone-healing treatment plants were obtained from the Indian Medicinal Plants, Phytochemistry and Therapeutics 2.0 (IMPPAT 2.0) (https://cb.imsc.res.in/imppat/), the

KNApSAcK family database (http://www.knapsackfamily.com/KNApSAcK/), and literature mining for thirty-two medicinal plants used as bone mending/healing by the traditional healers of Sikkim. The bioactive ligands were further screened for drug-likeness using MolSoft ProP software and SWISS-ADME bioavailability scores. Compounds with acceptable drug-likeness and a SwissADME bioavailability score greater than 0.4 were prioritised. The ligands that passed drug-likeness, bioavailability, and toxicity clearance screening were selected for further target prediction [11–14]. The Lipinski rule of five states that a compound with H-bond acceptor <10, H-bond donor <5, LogP<5, and molecular weight<500D attains good bioavailability. Thus, based on ADMET data, 249 compounds passed the Lipinski rule of five and 304 failed, out of 553 compounds (refer to Supplementary S3 and S4 **Tables**). However, some molecules that were often used but did not pass ADMET screening were further considered based on clinical and pharmacological evidence.

   **Target identification.**  The SMILES of the bioactive constituents were further searched in the SwissTargetPrediction, STITCH, and SuperPred databases [12,13]. Only targets with a prediction probability ≤0.6 (high-confidence interactions) were retained for further analysis to minimise false-positive associations. This multi-database consensus approach was adopted to enhance the reliability of target identification.

### Disease genes prediction

The GeneCard (7589), DisGeNET (90), and OMIM (66) databases were searched using the keyword "bone healing." Duplicated entries were removed, and the common genes were subjected to building a network [12,13]. A total of 707 overlapping genes were identified and used for network construction (Supplementary Tables 2 and 4).

### Protein-protein interaction network

STRING database used to construct disease gene network. A medium confidence interaction score (≥0.4) was applied to balance sensitivity and specificity of predicted interactions. Network topological parameters were analysed using Cytoscape, and hub genes were identified based on degree centrality. Nodes with degree values greater than 20 were considered key hub genes, indicating their potential regulatory importance within the network. KEGG pathway annotations were selected to build the PPI network [12,13]. Pathways and biological processes with a significance threshold of p<0.05 and fold enrichment >1.5 were considered statistically meaningful. These criteria enabled prioritisation of biologically relevant pathways involved in osteogenesis, inflammation, and bone remodelling.

## Results and discussion

### Literature validation

The local names of plants obtained from the ethnopharmacological survey were validated scientifically and authenticated by the Botanical Survey of India (BSI) – Regional Himalayan Centre, Government of India, Gangtok, Sikkim. The scientific names of the identified plants were verified on the Plants of the World Online website (https://powo.science.kew.org/) to ensure accuracy. A cross-referencing process was conducted to assess whether the plant cited in our study has been reported in scientific literature for the treatment of fractures, sprains, or related disorders in Sikkim, India. Furthermore, the scientific names were searched in PubMed, Google Scholar, and ScienceDirect using the terms "fracture, inflammation, orthopaedic trauma and sprain." These databases were also used to gather information on other ethnobotanical applications, to report the presence of bioactive phytoconstituents, and the pharmacological activity of the plants as revealed by the healers. This information is presented in a supplementary file (S1 Table).

### Informants

All respondents were interviewed about their histories and experiences with bone-healing practices. Most respondents were male (92.30%), reflecting male predominance in the healing practice. The age of the informant's healers (regardless

of gender) was all above 40 years old, with the largest proportion (53.84%) aged 60–70 years. Most healers (76.92%) had extensive experience, with over 20 years of experience (Table 1).

## The ingredients and traditional formulation

Based on a field survey, a total of 26 traditional bone-setters were interviewed, of whom 18 were willing to provide information on the formulations and recipes used for bone mending (Table 2). The traditional knowledge and treatment procedure of bone setting are generally passed down through generations, with personal experiences and observations often complementing this knowledge. Healing of a bone fracture is typically a natural, proliferative physiological process. The plant-based traditional practices primarily enhance cellular bone proliferation processes, promoting bone healing.

It has been noted that traditional bone-setters in Sikkim essentially use the same mending practices, which involve the following steps. Healers conduct initial examinations, and sometimes X-ray images are used to identify the type of fracture or sprain. This is followed by a gentle massage, as necessary, and cleaning of infected areas with commercial disinfectants.

Informants relocated the dislocated bone to normal and immobilised the fracture using a bamboo craft or stick. Traditional formulations, applied as a paste or as cinerite, were then placed on the fracture area and secured with cotton and bamboo. The findings of the present study are consistent with those of a previous report by Toafode et al. (2022), which described a similar treatment protocol for bone fractures and sprains in the northern Republic of Benin [16]. Moreover, 66.66% of healers supplement treatment with oral formulations.

The oral formulation includes herbal medicines and various adjuvants. Treatments are applied once or twice a week, and recovery time for mending varies, generally lasting 2–4 weeks depending on severity (Table 2). The medicinal plants used by traditional healers in Sikkim were systematically tabulated according to the formulations employed by each healer to identify plants that are frequently or commonly used. This frequency was the criterion for selecting prioritised medicinal plants or formulation components for further research and scientific validation (Fig 2).

The ethnopharmacological survey reported that 32 plant species, representing 20 families, were used orally, topically, or both to treat bone fractures (Table 3). The most frequently cited plant among recorded species is *Viscum articulatum* with a FoC of 16.11% and RFC of 0.77. Similarly, FoC for other species was as follows: *Kaempferia rotunda* 11.50%, *Astilbe rivularis* 10.30%, *Bergenia ciliata* 9.20%, *Fraxinus floribunda* 6.90%, *Euphorbia hirta* 6.90%, *Prunus cerasoides* 3.40%, *Macropanax dispermus* 3.40%, *Urtica parviflora* 2.30%, *Saurauia napaulensis* 2.30%, *Rubus ellipticus* 2.30%, and *Lepidium sativum* 2.30%. For all remaining species, the FoC was 1.10% (Fig 2). It is noted that a large number of

**Table 1. Demographic details of the traditional healers of Sikkim shared Traditional knowledge.**

| Variable | Categories | Frequency | Percentage |
|---|---|---|---|
| **Gender** | Male | 24 | 92.30 |
| | Female | 2 | 7.69 |
| **Age group** | <40 | NA | NA |
| | 40-50 | 3 | 11.53 |
| | 50-60 | 4 | 15.38 |
| | 60-70 | 13 | 53.84 |
| | >70 | 6 | 23.07 |
| **Experience** | < 2 | NA | NA |
| | 2-10 | 1 | 3.84 |
| | 10-20 | 5 | 19.23 |
| | > 20 | 20 | 76.92 |

**Table 2. The ingredients and traditional formulations used by traditional healers of Sikkim for treating bone fractures.**

| Formulation Code | Composition of Formulation | | Adjuvant Used |
|---|---|---|---|
| | **Topical** | **Oral** | |
| IBSD/ SK/F-1 | Boil 50g each of *K. rotunda* root and *L. sativum* WP for 30 min. Mix 50g each of *V. articulatum* WP and *L. sativum* WP with the boiled mixture to form a paste. Apply to the fracture, secure with a bamboo splint, and replace weekly for 3 weeks. | Boil WP of *V. articulatum* WP and *L. sativum* and take with honey single dose per day for 15 days | Honey |
| IBSD/ SK/F-2 | Crush and boil 50g each of *D. indica* bark, *F. floribunda*, *S. nepalensis* root, and *E. hirta* in 2 litres of water into a paste. Cool to room temperature, apply to the area, and secure with a bamboo splint. Change plaster twice weekly for 21 days. | Mix 5g each of *V. articulatum*, *L. sativum, A. rivularis, and B. ciliata* powders with an egg, 10 ml of cow milk, and 5g of honey. Take 3 doses every other day for 3 weeks. | Chicken egg, Cow milk and Honey |
| IBSD/ SK/F-3 | Boil the AP of *V. articulatim* to make a paste, apply it to the infected area, and secure it with a bamboo splint for 15 days. | Crush 3g *V. articulatim* into fine powder, boil with 250 ml water, and take 3 times daily for 15 days. | |
| IBSD/ SK/F-4 | Boil 100g each of *V. articulatum* WP*, S. wallichii* bark*, E. hirta* and *U. parvifora* roots in 500 mL of water into a paste. Apply to the area, bandage with cotton, support with a bamboo splint, and leave for 15–18 days. Reapply once if needed. | Crush *V. articulatim* into a fine powder, mix with 100 ml of Milk, 2.5 ml of Honey, and 2.5 ml of mustard oil, and take orally for 5 days. | Milk, Mustard oil, Honey |
| IBSD/ SK/F-5 | Crush 150g each of *V. articulatum* WP, *E. hirta*, and *K. rotunda* root with water into a paste. Apply and bandage with a bamboo splint for 15 days. Repeat once if severe. | Take 10g of powdered *B. ciliata, A. rivularis, K. rotunda* and *V. articulatum* daily with 100 ml milk and 2.5 ml honey. | Milk and Honey |
| IBSD/ SK/F-6 | Mix 20g each of *C. infertunatum WP and K. rotunda* rhizome into a paste, apply, bandage, and secure with a bamboo splint for 15 days. | ND | ND |
| IBSD/ SK/F-7 | Crush 50g *A. Manihot* root, 25g *M. rubicaulis* root, 50g *K. rotunda* rhizome, and 25g *L. Sativum* seed into a paste, apply and secure with a bamboo splint for 15 days. | Crush 20g each of *A. rivularis, B. ciliata* and 30g WP of *V. articulatim* and take 5 grams in 100 ml water for 10 days. | |
| IBSD/ SK/F-8 | Combine 50g each of *V. articulatum* WP, *U. parvifora* roots, *E.* hirta, *F. floribunda* bark, and *K. rotunda* rhizome and crush into a paste. Apply, bandage, and brace with bamboo for 15 days. | Take 10g of *F. floribunda* bark, *A. rivularis, B. ciliata, R. calycnus*, and *V. articulatum* WP powder with 125 ml water daily. | |
| IBSD/ SK/F-9 | Crush 50g each of *C. umbrosum*, *K. rotunda*, and *E. hirta* roots into a paste, apply to the area, bandage securely with a bamboo splint, and change weekly thrice. | Take 2.5g of powdered *V. articulatum* WP, *C. umbrosum, B. ciliata* roots*, A. rivularis*, and *F. elastica* bark with 125 ml milk and 2.5 ml honey, twice daily until healed. | Honey and milk |
| IBSD/ SK/F-10 | Crush 500g *P. cerasoides* bark in 5 L of water, filter, boil to a paste, apply, bandage with a bamboo splint, and leave for 9 days. | Soak 100g each of *A. rivularis* and *B. ciliata* roots in 500 ml of water overnight; take 20 ml twice daily. | ND |
| IBSD/ SK/F-11 | Crush 100g each of *F. floribunda* bark, *D. indica, K. rotunda, G. diversifolia* roots, and *V. articulatum* WP; form a paste, apply on the fracture, bandage with a bamboo splint, and leave for 5 days. | ND | ND |
| IBSD/ SK/F-12 | Boil 100g each of *F. floribunda* bark, *D. indica, V. articulatum* WP, and *K. rotunda* rhizome in 1.5 L water into a paste. Apply to the fracture, bandage, and leave for one week. | | |
| IBSD/ SK/F-13 | Boil and filter 50g each of *S. nepalensis, F. bengalensis, F. vierns, P. cerasoides, F. floribunda* barks, and *V. articulatum* WP. Dry the filtrate, make a paste, apply it to the fracture, bandage with a bamboo splint, change weekly, repeat thrice. | Powder 100g each of *V. articulatum* WP, *A. lakoocha, B. alnoides* barks, *S. album* root, *A. rivularis*, and *H. wallichii* flower. Boil 500 ml of water until reduced by half. Take 5 ml twice daily for 7 days. | |
| IBSD/ SK/F-14 | Boil 100g each of *E. spicata, E. meliaefolia* barks, *P. cerasoides*, and *C. sativus* root in 1.5 L of water into a paste. Apply to the fracture, bandage with a bamboo splint, and leave for 30 days. Treatment is applied once a week. | | |

Note: Composition of all formulations mentioned in the table as per the traditional knowledge shared by traditional healers.

| Traditional Formulation | Plants used for traditional formulation (serial number 1-32 corresponds to the respective plants listed below) | | | | | | | | | | | | | | | | | | | | | | | | | | | | | | | |
|---|---|---|---|---|---|---|---|---|---|---|---|---|---|---|---|---|---|---|---|---|---|---|---|---|---|---|---|---|---|---|---|---|
| | 1 | 2 | 3 | 4 | 5 | 6 | 7 | 8 | 9 | 10 | 11 | 12 | 13 | 14 | 15 | 16 | 17 | 18 | 19 | 20 | 21 | 22 | 23 | 24 | 25 | 26 | 27 | 28 | 29 | 30 | 31 | 32 |
| IBSD/SK/F-1 | Y | Y | | | | | | | | | | Y | | | | | | | | | | | | | | | | | | | | |
| IBSD/SK/F-2 | Y | | Y | Y | Y | Y | | Y | | Y | | Y | | | | | | | | | | | | | | | | | | | | |
| IBSD/SK/F-3 | Y | | | | | | | | | | | | | | | | | | | | | | | | | | | | | | | |
| IBSD/SK/F-4 | Y | | | | | Y | | | Y | | | | | | | | | | | Y | | | | | | | | | | | | |
| IBSD/SK/F-5 | Y | Y | Y | Y | | Y | | | | | | | | | | | | | | | | | | | | | | | | | | |
| IBSD/SK/F-6 | | Y | | | | | | | | | | | | | | | | | | | | | | | | Y | | | | | | |
| IBSD/SK/F-7 | Y | Y | Y | Y | | | | | | | | | Y | | | | | | | | | | | | | | | Y | | | | |
| IBSD/SK/F-8 | Y | Y | Y | Y | Y | Y | | | Y | | Y | | | | | | | | | | | | | | | | | | | | | |
| IBSD/SK/F-9 | Y | Y | Y | Y | | Y | | | | | | | | | | | Y | | | | | | | | | | | | | | Y | |
| IBSD/SK/F-10 | | Y | Y | | | | Y | | | | | | | | | | | | | | | | | | | | | | | | | |
| IBSD/SK/F-11 | Y | Y | | | Y | | | Y | | | | | | | Y | | | | | | | | | | | | | | | | | |
| IBSD/SK/F-12 | Y | Y | | | Y | | | Y | | | | | | | | | | | | | | | | | | | | | | | | |
| IBSD/SK/F-13 | Y | | Y | | Y | | Y | | | Y | | | | Y | | | | Y | | | Y | | | Y | | | | | Y | | | Y |
| IBSD/SK/F-14 | | | | | | | Y | | | | | | | | | Y | | | | | | | | | Y | | Y | | | | | |
| IBSD/SK/F-15 | Y | | | | | | | | | | | | | | | | | | | | | Y | | | | | | | | Y | | |
| IBSD/SK/F-16 | Y | Y | Y | Y | Y | Y | | | | | | | | | | | | | | | | | | | | | | | | | | |
| IBSD/SK/F-17 | | Y | | | | | | | | | | | | | | | | | Y | | | | Y | | | | | | | | | |
| IBSD/SK/F-18 | Y | | Y | Y | | | | | | | Y | | | | | | | | | | | | | | | | | | | | | |
| Total-18 | 14 | 10 | 09 | 08 | 06 | 06 | 03 | 03 | 02 | 02 | 02 | 02 | 01 | 01 | 01 | 01 | 01 | 01 | 01 | 01 | 01 | 01 | 01 | 01 | 01 | 01 | 01 | 01 | 01 | 01 | 01 | 01 |
| Frequency of citation | 16.11 | 11.5 | 10.3 | 09.2 | 06.9 | 06.9 | 03.4 | 03.4 | 02.3 | 02.3 | 02.3 | 02.3 | 01.1 | 01.1 | 01.1 | 01.1 | 01.1 | 01.1 | 01.1 | 01.1 | 01.1 | 01.1 | 01.1 | 01.1 | 01.1 | 01.1 | 01.1 | 01.1 | 01.1 | 01.1 | 01.1 | 01.1 |

*1. Viscum articulatum, 2. Kaempferia rotunda, 3. Astilbe rivularis, 4. Bergenia ciliata, 5. Fraxinus floribunda, 6. Euphorbia hirta, 7. Prunus cerasoides, 8. Macropanax dispermus, 9. Urtica parviflora, 10. Saurauia napaulensis, 11. Rubus ellipticus, 12. Mimosa rubicaulis, 13. Artocarpus lakoocha, 14. Girardinia diversifolia, 15. Trichosanthes cucumerina, 16. Clinopodium umbrosum, 17. Ficus benghalensis, 18. Lepidium Sativum, 19. Heracleum wallichii, 20. Schima wallichii, 21. Zingiber montanum, 22. Centella asiatica, 23. Ficus virens, 24. Curcuma caesia, 25. Clerodendrum infortunatum, 26. Abelmoschus manihot, 27. Euodia meliifolia, 28. Engelhardia spicata, 29. Santalum album, 30. Rheum nobile, 31. Ficus elastica, 32. Betula alnoides .*

**Fig 2. The criterion for selection of prioritised medicinal plants or components of the formulation for further research and scientific validation.**

species belonged to the Moraceae family, followed by the Zingiberaceae family. There were two species each of Polygonaceae, Santalaceae, Lamiaceae, Apiaceae, Rosaceae, Urticaceae, and Saxifragaceae, and one species each from the Oleaceae, Euphorbiaceae, Actinidiaceae, Fabaceae, Brassicaceae, Theaceae, Malvaceae, Rutaceae, Juglandaceae, and Betulaceae. Additionally, it was noted that among the plant parts used, bark was most frequently used at 40.62%, followed by root at 21.87%, rhizome at 15.62%, and flowers at 3.14%. Whole plants were used for the six species (18.75%). Herbs are more frequently reported in the formulations (44%), followed by trees (37%) and shrubs (19%). Bark was most commonly used because it is a collection of wood and the dead tissues of vascular plants, mainly composed of lignin, polyphenols, suberin, and organic and inorganic minerals. The plant bark typically contains 3–17% of organic compounds and 1.5–10.0% of inorganic minerals, predominantly Nitrogen, Calcium, and Potassium [17]. The presence of various phytochemicals and minerals enhances the chemotherapeutic potential, particularly in treating bone fractures and orthopaedic trauma.

In addition to the medicinal plants, various adjuvants were used in the formulations. Among the adjuvants, honey is most commonly used in both oral and topical preparations. Honey has been used to treat several conditions, including wounds, cataracts, burns, bone remodelling, and osteoporosis. Natural honey has demonstrated antioxidant and

**Table 3. Plants used by Traditional healers of Sikkim for bone mending.**

| Plants Taxa | Voucher No | Plant Family | Local name (Nepali) | Plant part used | Mode of Preparation | Mode of Application | Frequency of citation (FoC) | Relative Frequency of Citation (RFC) | Therapeutic Uses/Use reports | ICPC Code (URs) |
|---|---|---|---|---|---|---|---|---|---|---|
| *Viscum articulatum* Burm. f. | IBSD-RC/EPS/22/NS/53 | Santalaceae | Harchur | Whole plant | Decoction, powder | Tropical & oral | 16.11 | 0.77 | Bone fracture | L74 (14) |
| *Kaempferia rotunda* L. | IBSD-RC/EPS/22/WS/55 | Zingiberaceae | Bhui Champa | Rhizome | Decoction, paste | Topical & oral | 11.50 | 0.55 | Bone fracture | L74 (10) |
| *Astilbe rivularis* Buch. -Ham. ex D.Don | IBSD-RC/EPS/22/WS/2 | Saxifragaceae | Buro-okhati | Rhizome | Decoction, powder | Topical & oral | 10.30 | 0.50 | Bone fracture | L74 (9) |
| *Bergenia ciliata* (Haw.) Sternb. | IBSD-RC/EPS/22/SS/33 | Saxifragaceae | Pakhenbed | Rhizome | Decoction, paste, powder | Topical & oral | 9.20 | 0.44 | Bone fracture | L74 (8) |
| *Fraxinus floribunda* Wall. | IBSD-RC/EPS/22/WS/6 | Oleaceae | Lakuri | bark | Decoction, paste, powder | Topical & oral | 6.90 | 0.33 | Bone fracture | L74 (6) |
| *Euphorbia hirta* L. | IBSD-RC/EPS/22/WS/8 | Euphorbiaceae | Bhui Chipley | whole plant | Decoction, paste | topical | 6.90 | 0.33 | Bone fracture | L74 (6) |
| *Prunus cerasoides* Buch.-Ham. ex D.Don | IBSD-RC/EPS/22/NS/58 | Rosaceae | Payum | bark | Decoction | topical | 3.40 | 0.16 | Bone fracture | L74 (3) |
| *Macropanax dispermus* (Wallich ex G. Don) | IBSD-RC/EPS/22/SS/57 | Araliaceae | Pachpatey | bark | Decoction, paste | topical | 3.40 | 0.16 | Bone fracture | L74 (3) |
| *Urtica parviflora* Roxb. | IBSD-RC/EPS/22/PK/33 | Urticaceae | Gharia sisnu | root | Decoction | topical | 2.30 | 0.11 | Bone fracture | L74 (2) |
| *Saurauia napaulensis* DC. | IBSD-RC/EPS/22/NS/48 | Actinidiaceae | Gagun | bark | Decoction, paste | topical | 2.30 | 0.11 | Bone fracture | L74 (2) |
| *Rubus ellipticus* Wall. ex D.Don | IBSD-RC/EPS/22/GT/3 | Rosaceae | Aishleo | root | powder | oral | 2.30 | 0.11 | Bone fracture | L74 (2) |
| *Lipidum sativum* L | IBSD-RC/EPS/22/NS/41 | Brassicaceae | Chausor | whole plant | Decoction, Powder | Topical & oral | 2.30 | 0.11 | Bone fracture | L74 (2) |
| *Mimosa rubicaulis* subsp. himalayana (Gamble) H.Ohashi | IBSD-RC/EPS/22/SS/52 | Fabaceae | Ararey | root | Decoction, paste | topical | 1.10 | 0.05 | Bone fracture | L74 (1) |
| *Artocarpus lakoocha* Roxb. | IBSD-RC/EPS/22/WS/61 | Moraceae | Barar | bark | Decoction | oral | 1.10 | 0.05 | Bone fracture | L74 (1) |
| *Girardinia diversifolia* (Link) Friis | IBSD-RC/EPS/22/NS/37 | Urticaceae | Bhagre sisnu | root | paste | topical | 1.10 | 0.05 | Bone fracture | L74 (1) |
| *Trichosanthes cucumerina* L. | IBSD-RC/EPS/22/SS/41 | Cucurbitaceae | Bhaishe shig | root | Decoction | topical | 1.10 | 0.05 | Bone fracture | L74 (1) |
| *Clinopodium umbrosum* (M.Bieb.) K.Koch | IBSD-RC/EPS/22/NS/52 | Lamiaceae | Billa Jhora | whole plant | Decoction, paste | Topical & oral | 1.10 | 0.05 | Bone fracture | L74 (1) |

*(Continued)*

**Table 3.** (Continued)

| Plants Taxa | Voucher No | Plant Family | Local name (Nepali) | Plant part used | Mode of Preparation | Mode of Application | Frequency of citation (FoC) | Relative Frequency of Citation (RFC) | Therapeutic Uses/Use reports | ICPC Code (URs) |
|---|---|---|---|---|---|---|---|---|---|---|
| *Ficus bengalensis* L. | IBSD-RC/EPS/22/NS/46 | Moraceae | Bor (patty) Seti bar | root | Decoction | topical | 1.10 | 0.05 | Bone fracture | L74 (1) |
| *Heracleum wallichii* DC. | IBSD-RC/EPS/22/NS/51 | Apiaceae | Chimping | bark, flower | Decoction | Topical & oral | 1.10 | 0.05 | Bone fracture | L74 (1) |
| *Schima wallichii* (DC.) Korth. | IBSD-RC/EPS/22/WS/65 | Theaceae | Chilaune | bark | Decoction | topical | 1.10 | 0.05 | Bone fracture | L74 (1) |
| *Zingiber montanum* (J. Koenig) Link ex A. Dietr. | IBSD-RC/EPS/22/NS/53 | Zingiberaceae | Fakchem | rhizome | paste | topical | 1.10 | 0.05 | Bone fracture | L74 (1) |
| *Centella asiatica* (L.) Urb. | IBSD-RC/EPS/22/WS/29 | Apiaceae | Golpatta | whole plant | paste | topical | 1.10 | 0.05 | Bone fracture | L74 (1) |
| *Ficus virens* Aiton. | IBSD-RC/EPS/22/NS/47 | Moraceae | Kabra (Ficus) | bark | decoction | topical | 1.10 | 0.05 | Bone fracture | L74 (1) |
| *Curcuma caesia* Roxb | IBSD-RC/EPS/23/WS/66 | Zingiberaceae | Kalo haldi | rhizome | paste | topical | 1.10 | 0.05 | Bone fracture | L74 (1) |
| *Clerodendrum infortunatum* L. | IBSD-RC/EPS/23/SS/55 | Lamiaceae | kalochito | whole plant | paste | topical | 1.10 | 0.05 | Bone fracture | L74 (1) |
| *Abelmoschus manihot* var. *pungens* (Roxb.) Hochr. | IBSD-RC/EPS/23/PK/51 | Malvaceae | kapasay/jangli vindi | root | paste, decoction | topical | 1.10 | 0.05 | Bone fracture | L74 (1) |
| *Euodia fraxinifolia(Hook) Hartley.* | IBSD-RC/EPS/23/NS/8 | Rutaceae | Khanakpa | fruit | Powder | oral | 1.10 | 0.05 | Bone fracture | L74 (1) |
| *Engelhardia spicata* Lesch. Ex Blume | IBSD-RC/EPS/22/NS/45 | Juglandaceae | Mauwa | bark | decoction | topical | 1.10 | 0.05 | Bone fracture | L74 (1) |
| *Santalum album* L. | IBSD-RC/EPS/WS/23/12 | Santalaceae | Molagiri | root | decoction | oral | 1.10 | 0.05 | Bone fracture | L74 (1) |
| *Rheum nobile* Hook.f. & Thomson | IBSD-RC/EPS/NS/23/51 | Polygonaceae | Padamchal | rhizome | Paste | topical | 1.10 | 0.05 | Bone fracture | L74 (1) |
| *Ficus elastica* Roxb. Ex Hornem | IBSD-RC/EPS/22/WS/67 | Moraceae | Labar | bark | Powder, Decoction | oral | 1.10 | 0.05 | Bone fracture | L74 (1) |
| *Betula alnoides* Buch.-Ham. Ex D.Don | IBSD-RC/EPS/23/SS/11 | Betulaceae | Saur | bark | Powder | oral | 1.10 | 0.05 | Bone fracture | L74 (1) |

Note: Scientific name of the medicinal plants given as per the nomenclatures listed in Plants of the World Online (POWO), https://powo.science.kew

anti-inflammatory properties, playing a crucial role in wound healing and fracture repair. The honey primarily consists of sugars, organic acids, ascorbic acids, flavonoids, phenols and natural enzymes [18]. The anti-inflammatory and antioxidant activities of honey are attributed to its phytoconstituents, mainly flavonoids and phenols. Furthermore, various studies reported that honey increases calcium absorption in experimental animals [19].

Moreover, others have reported honey's fracture-healing properties through bone union, new bone formation, and bone remodelling [20,21]. Mustard oil is commonly used for massaging the fracture areas. Mustard is a commonly grown oilseed crop belonging to the Brassicaceae family. Glucosinolase, sinalbin, sinapic acid, eicosenoic, arachidic, nonadecanoic, behenic, oleic and palmitic acids and sinapine are the principal chemical components of mustard [22]. The composition of various bioactive constituents makes it a potential anti-inflammatory and antioxidative agent [23].

Shilajit is a multicomponent, naturally occurring mineral exudate extracted from the layer of mountainous rocks [3]. Shilajit has been ascribed a role in mitigating various biological ailments. The Shilajit mainly comprises humus and other organic constituents, including fulvic acid (60% to 80%) and multiple oligo-elements [24,25]. Shilajit has been used for treating bone fractures in traditional Persian Medicine (TPM) [26]. Moreover, it effectively maintains bone health [27]. Similarly, Kacho Simrik is a brick-red, multicomponent, mineralised compound extracted from rocks. It has been used to treat various ailments, particularly women's menstrual problems, and for wound healing and bone fractures. Furthermore, Cow milk and Eggs are natural sources of protein that help maintain bone health [28,29].

## Literature review and biological activity of cited plant species

The present study suggests that various plant species may have significant potential for bone repair. A review of the literature on the species most frequently used by traditional healers for bone mending is provided in a supplementary file (S1 Table).

*Viscum articulatum* is a hemiparasitic plant belonging to the Viscaceae family [30]. *Viscum articulatum* possesses a wide range of medicinal properties and has been utilised ethnobotanically and traditionally by various communities across India, Nepal and China to treat various ailments, including inflammation, bone fractures, joint pain and wound treatment [31].

Moreover, *Viscum articulatum* has also been previously reported for treating bone fractures in West Sikkim. Different parts of *Viscum articulatum* reveal the presence of various phytochemicals like lectins, viscotoxins, flavonoids, terpenoids, glycosides, phenolic compounds, steroids, tannins and polysaccharides [32]. In addition, five terpenoids, including a-amyrin, botulin, betulinic acid, lupeol and oleanolic acid, were also reported to be present in the plant [4,31].

However, the RP-HPLC analysis reveals that the methanolic extract of *Viscum articulatum* contains significant amounts of oleanolic acid [33]. Interestingly, there is some evidence suggesting that oleanolic acid exhibits osteoblastic and bone-protective activity [34]. Therefore, the presence of bioactive phytochemicals, including tannins, in *Viscum articulatum* may help reduce inflammation and promote bone-healing activity.

*Kaempferia rotunda* is a perennial medicinal herb of the Zingiberaceae family. The rhizomes of *Kaempferia rotunda* have been reported from Tripura for use in traditional practices to treat bone fractures [35]. In addition to bone healing, it also exhibits antioxidant and anti-inflammatory activity [36]. Furthermore, the reported presence of various major phytochemicals, such as benzyl benzoate, n-pentadecane, Camphene, Camphor, β-pinene, α-pinene and Linalool oxides, may contribute to bone-mending activity [37]. α- and β-pinene promote osteoclastic activity, thereby initiating normal bone remodelling [38].

*Astilbe rivularis* is a rhizomatous herb belonging to the Saxifragaceae family. The paste of *Astilbe rivularis* is used for sprains, muscular swelling, bone fractures and joint dislocations. The rhizome is also used to treat diarrhoea, peptic ulcer and dysentery [39]. *Astilbe rivularis* has been reported to contain various bioactive compounds, including Coumarins, aesculetin, astilbic acid, astilbin, aticoside, dimethylaesculetin, daucosterol, eucryphin, palmitine, peltoboykinoleic acid, scopoletin, sitosterol, stilbene, Bergenin and bergenin derivatives, β-amyrin, β-sitosterol and Butanedioic acid [39–41]. Daucosterol is a novel bioactive compound that promotes osteoblastic cell proliferation [42].

*Bergenia ciliata* is an herbaceous perennial from the Saxifragaceae family that has been employed to treat pulmonary infections, piles, dysuria, ulcers and stones in the kidney and gallbladder [43]. The local community in India used rhizomes to treat bone fractures, cuts, and wounds [44]. Kour et al. (2021) reported that the rhizome ethanolic extract of *Bergenia ciliata* improves wound healing in experimental animals [45]. Bergenin is a major bioactive phytoconstituent present in the rhizome of *Bergenia ciliata* [5]. Bergenin promotes bone fracture healing by upregulating the SIRT1 gene and promotes osteogenic differentiation of bone mesenchymal stem cells [46].

*Fraxinus floribunda* is a large, green tree belonging to the Oleaceae family, found in the eastern Himalaya of India. The bark of the plant possesses antioxidant, anti-inflammatory, anti-nociceptive and anti-arthritic activity [47–49]. Previous studies have reported the use of the bark to treat bone fractures, bone dislocations, and gout by traditional healers from Sikkim [50]. The presence of phytoconstituents such as 8-acetyl-7-hydroxy-6-methoxycoumarin, 8-methoxycoumarin, 2,5-dihydroxy-6-methoxyacetophenone, fraxetin and esculetin may help maintain bone health and treat fractures. In addition, studies have shown that esculetin increases bone mass by upregulating RANKL expression [51].

*Euphorbia hirta* is a slender-stemmed, hairy plant from the Euphorbiaceae family. The plant has been reported to exhibit various pharmacological activities: leaf extracts show antimicrobial, wound-healing, and burn-healing activities, while the aerial part shows anti-inflammatory, analgesic, antipyretic, antihistaminic, hepatoprotective, and anticancer activities [52]. Moreover, the crushed roots of *Euphorbia hirta* have been used to treat bone fractures in the Ranchi district of Jharkhand [53]. Afzelin, rutin, quercitin, euphorbin-A, kaempferol, gallic acid, β-amyrin, β-sitosterol, chtolphenolic acid, chlorophenolic acid, leucocyanidin, myricitrin, cyaniding 3,5-diglucoside, camphol, flavanol, inositol and tetraxerol are the major phytoconstituents present in the *Euphorbia hirta* [54–56]. Rutin, quercetin, kaempferol, and other dietary flavonoids are reported to have osteoprotective activity in the treatment of bone fractures [57–59].

It has been observed that all plants contain bioactive compounds such as terpenoids, alkaloids and flavonoids, which can contribute significantly to the treatment of bone fractures, sprains, wounds and inflammations [60]. Several plants reported in the present study, like *Viscum articulatum, Kaempferia rotunda, Astilbe rivularis, Fraxinus floribunda, Euphorbia hirta, Bergenia ciliata, Prunus cerasoides, Lepidium sativum, Mimosa rubicaulis*, *Artocarpus lakoocha, Ficus benghalensis, Heracleum wallichii, Schima wallichii, Curcuma caesia, Abelmoschus manihot, Rheum nobile,* and *Ficus elastica,* were reported to possess anti-inflammatory activity. These plants may therefore serve as potential candidates for pharmacological studies focused on bone-healing activity. Additionally, the phytochemical constituents of several species, including *Urtica parviflora, Ssaurauia napaulensis*, *Rubus calycinus,* and *Schima wallichii*, have not been examined. Comprehensive investigations into the phytochemicals and their pharmacological properties are essential.

Present studies enhance traditional treatment approaches by providing valuable insights into the pharmacological activity of these compounds. Furthermore, these inquiries may culminate in the development of plant-derived herbal formulations with demonstrated clinical efficacy and confirmed safety for bone healing.

Although increased exposure to modern medicines has eroded some traditional healthcare practices, the local community continues to rely primarily on traditional treatment methods, especially for fractures and sprains. There were no complaints about the efficacy and safety of the traditional herbal formulations among the locals who received the treatment, providing anecdotal support for the effectiveness of local treatment methods. Nevertheless, value-added research is necessary to validate the traditional healthcare system scientifically and to help develop low-cost therapeutics for treating bone fractures.

## Network annotations

A total of 1,043 phytocompounds were initially identified, of which 553 redundant compounds were discarded, and 527 were selected for further annotations (Supplementary S2 Table). Target prediction using the Swiss Target prediction database yielded 15,949 target genes (Supplementary material S3 Table), and after removing redundancy, 7,612 genes associated with bone healing were retrieved from the GeneCard, DisGeNET, and OMIM databases (Fig 3,4). A total of 707 common genes were used to obtain various pathway interactions and a protein-protein interaction network. Pathway

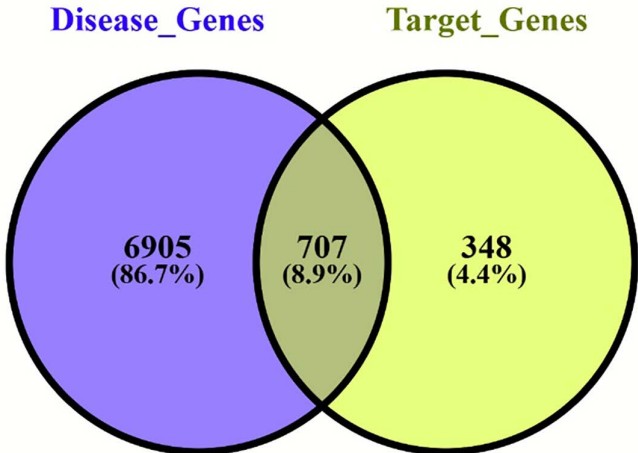

**Fig 3. Venn diagram depicting the target genes and disease genes obtained from Swiss Target Prediction, STITCH, Superpred, GeneCard, DisGeNET and OMIM, respectively.** The large target dataset was refined to p-values> 0.6.

enrichment analysis identified eleven significantly enriched pathways predominantly affecting bone healing. These included pathways governs biomineralisation, bone development, calcium signalling pathway (hsa04020), endochondral ossification with skeletal dysplasias (WP4808), non-genomic actions of 1,25-dihydroxyvitamin D3 (WP4341), oncostatin M signalling (WP2374), osteoclast differentiation (hsa04380) (Fig 5), RANKL-RANK signalling (WP2018), RUNX2 regulates bone development (HSA-8941326), RUNX2 regulates osteoblast differentiation (HSA-8940973), and Vitamin D-sensitive calcium signalling in depression (WP4698). Collectively, these pathways converge upon processes critical for bone homeostasis, including calcium regulation, osteoblast maturation, and osteoclast-mediated remodelling. Biological Process, Reactome Enrichment Pathways and selected protein interactions of respective pathways are highlighted in the supplementary material (S1–S19 Fig).

The protein-protein interaction network is dense and shows predominant gene interactions, especially with TERT, SRC, CYP27B1, ESR1, FLT3, AKT, and MM13 (Fig 4, right panel). These hub proteins are mostly involved in cellular survival and tissue repair mechanisms. Notably, CYP27B1 plays a central role in systemic calcium homeostasis. This enzyme catalyses the conversion of 25-hydroxyvitamin D to its active form, 1,25-dihydroxyvitamin D, in the renal proximal tubule, potentially influencing the renal production of active vitamin D. The compounds targeting CYP27B1 may enhance intestinal calcium absorption and maintain calcium availability for bone mineralisation. Additionally, SRC integrates calcium-dependent signals downstream of the calcium-sensing receptor (CaSR) and RANKL-mediated pathways in osteoclasts and osteoblasts. The SRC signalling regulates cytoskeletal organisation through intracellular calcium fluxes. The dual target availability for CYP27B1 and SRC reflects a multilevel regulatory network that may help in bone healing. These predicted interactions offer a theoretical framework of traditional formulation and its pleiotropic effects on calcium metabolism; however, it is important to note that these findings represent in silico prediction and warrant experimental validation

## Conclusion and future directions

Sikkim is home to rich cultural diversity and diverse traditional healthcare practices and rituals. Traditional formulations for treating bone fractures are prevalent in rural areas of Sikkim. The present study documented 18 traditional polyherbal formulations utilising 32 medicinal plants for the treatment of bone fractures in Sikkim. In the realm of ethnobotany, the utilisation of quantitative indices not only facilitates the comparison and analysis of ethnobotanical data but also enables researchers to derive meaningful insights into the intricate connections between human societies and plant resources.

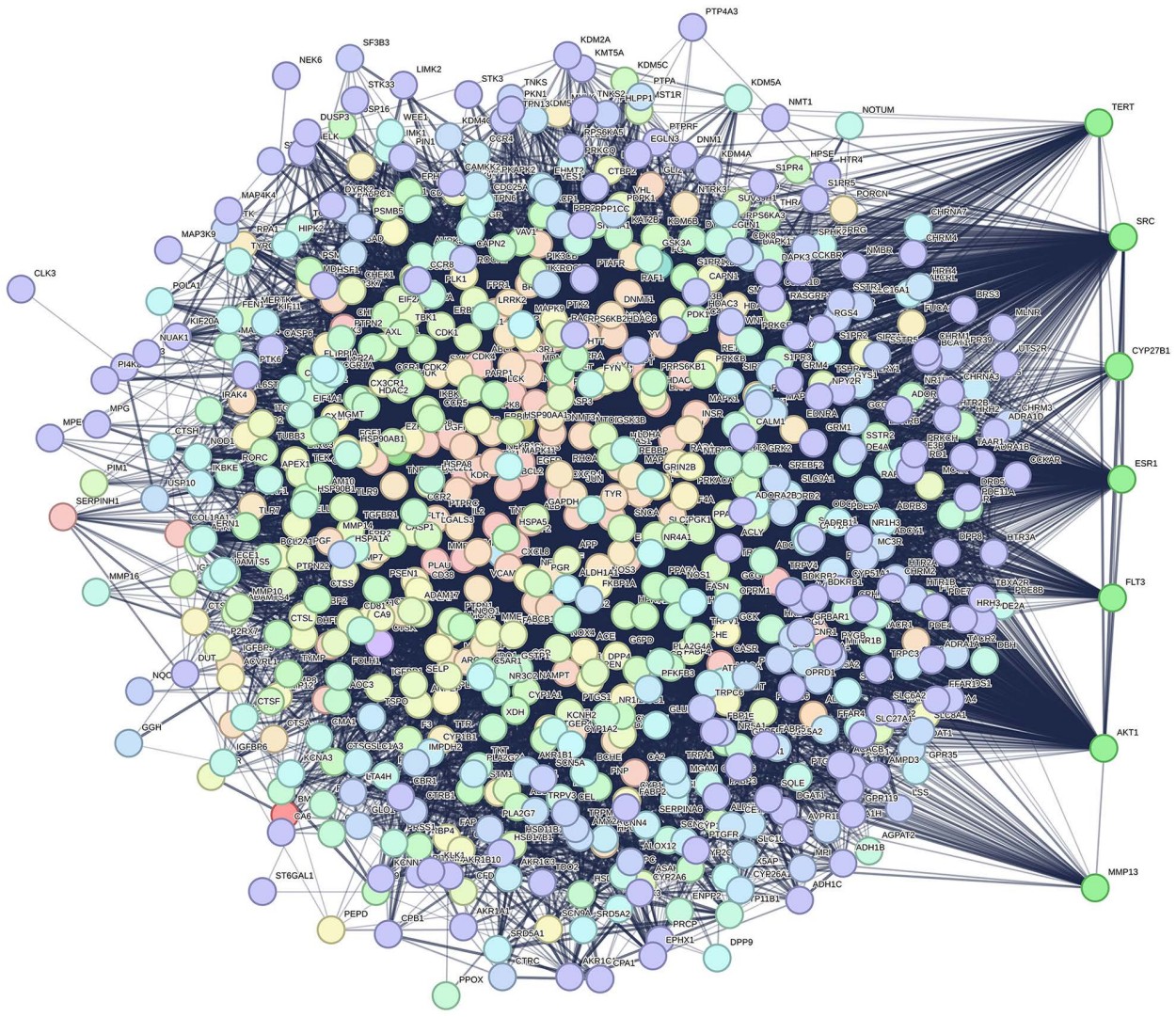

**Fig 4. Protein-protein interaction network of genes associated with bone mending.** The list of common genes with dense networks is shown on the right side of the figure (TERT, SRC, CYP27B1, ESR1, FLT3, AKT, and MM13).

The study resulted in documentation that provides an overview from a phytopharmacological perspective, as well as other pharmacological activities and modes of application for treating bone fractures and trauma. The recorded plants are utilised in both topical applications and oral formulations to alleviate pain, swelling and inflammation, and to promote bone healing. The present documentation can provide substantial support for the recorded plants, especially those with high FoC, for phytochemical and clinical assessments.

This analysis sheds light on the significance and recognition these plants have garnered within the research community. Furthermore, the direct and indirect associations of bioactive compounds affecting bone-mending pathways are supported by network pharmacological annotations. It also emphasises the importance of further exploration of the characteristics and properties of these plants in future studies. Moreover, a proper scientific study of plant phytochemistry is recommended to develop therapeutics for bone fractures.

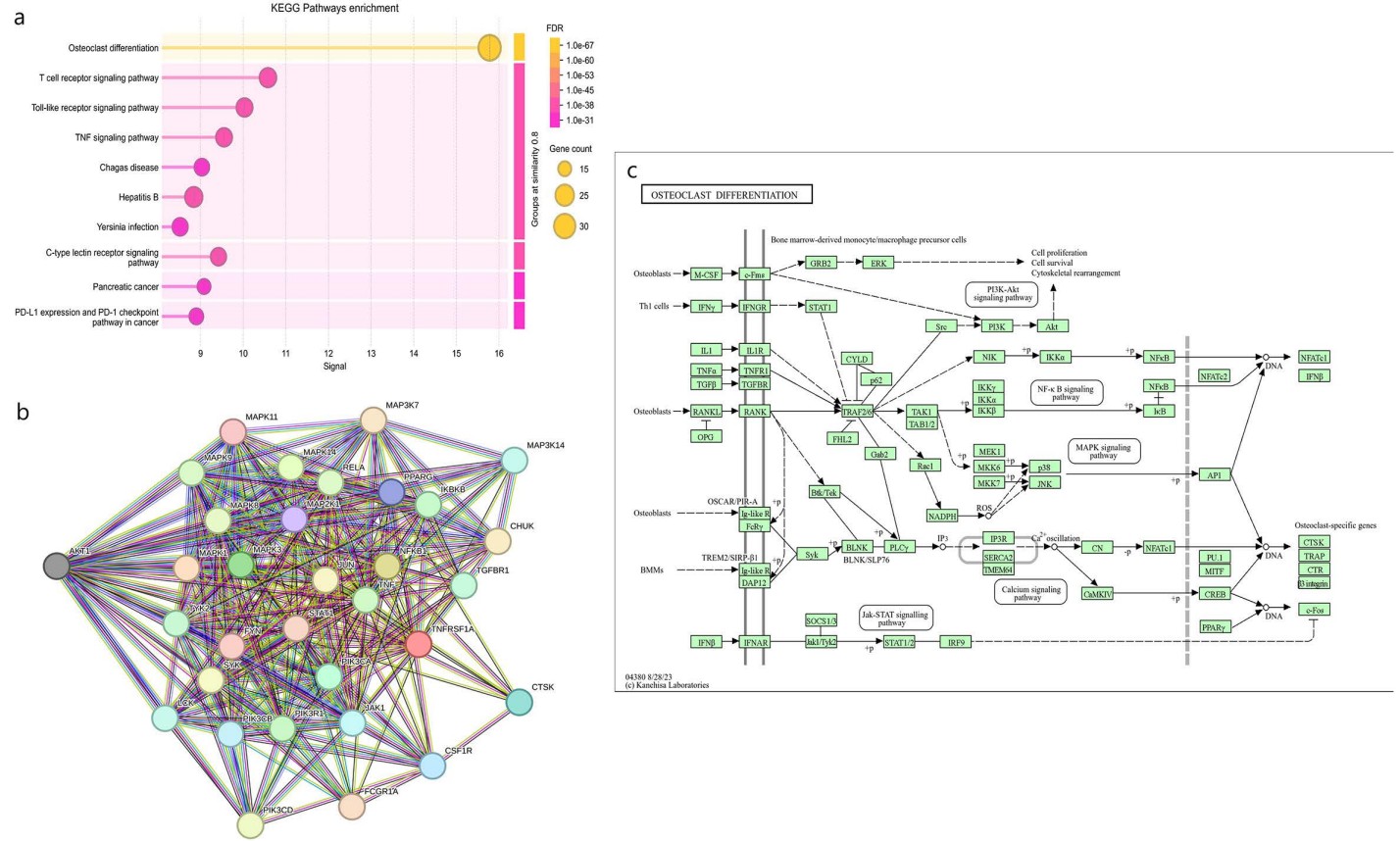

**Fig 5. Osteoclast differentiation (a) KEGG Pathways Enrichment, (b) PPI of selective genes and (c) Biological Pathway.**

## Supporting information

**S1 File. Methodology.** Methodology of ethnopharmacological survey and documentation with attached questionnaires.
(DOCX)

**S1 Fig. Biomineralization pathway (a) Biological Process, (b) Molecular Function and (c) Reactome Pathways Enrichment.**
(JPG)

**S2 Fig. Biomineralization pathway (a) Wiki Pathways Enrichment, (b) PPI of selective genes.**
(JPG)

**S3 Fig. Bone Development (a) Biological Process, (b) Molecular Function and (c) Cellular Component.**
(JPG)

**S4 Fig. Bone Development (a) Wiki Pathways Enrichment, (b) PPI of selective genes.**
(JPG)

**S5 Fig. Calcium Signaling Pathway (a) Biological Process, (b) Molecular Function and (c) Cellular Component.**
(JPG)

**S6 Fig. Calcium Signaling Pathway (a) KEGG Pathways Enrichment, (b) PPI of selective genes and (c) Biological Pathway.**
(JPG)

**S7 Fig. Endochondral Ossification with Skeletal Dysplasia (a) Biological Process, (b) Molecular Function and (c) Cellular Component.**
(JPG)

**S8 Fig. Endochondral Ossification with Skeletal Dysplasia (a) Wiki Pathways Enrichment, (b) PPI of selective genes and (c) Biological Pathway.**
(JPG)

**S9 Fig. Non-genomic actions of 1,25-dihydroxyvitamin D3 (a) Biological Process, (b) Molecular Function and (c) Cellular Component.**
(JPG)

**S10 Fig. Non-genomic actions of 1,25-dihydroxyvitamin D3 (a) Wiki Pathways Enrichment, (b) PPI of selective genes and (c) Biological Pathway.**
(JPG)

**S11 Fig. Oncostatin M signaling (a) Biological Process, (b) Molecular Function and (c) Cellular Component.**
(JPG)

**S12 Fig. Oncostatin M signaling (a) Wiki Pathways Enrichment, (b) PPI of selective genes and (c) Biological Pathway.**
(JPG)

**S13 Fig. Osteoclast differentiation (a) Biological Process, (b) Molecular Function and (c) Cellular Component.**
(JPG)

**S14 Fig. Osteoclast differentiation (a) KEGG Pathways Enrichment, (b) PPI of selective genes and (c) Biological Pathway.**
(JPG)

**S15 Fig. RANKL/RANK signaling (a) Biological Process, (b) Molecular Function and (c) Cellular Component.**
(JPG)

**S16 Fig. RANKL/RANK signaling (a) Wiki Pathways Enrichment, (b) PPI of selective genes and (c) Biological Pathway.**
(JPG)

**S17 Fig. RUNX2 regulates bone development (a) Biological Process, (b) Molecular Function and (c) Cellular Component.**
(JPG)

**S18Fig. RUNX2 regulates bone development (a) Reactome Pathways Enrichment, (b) PPI of selective genes and (c) Biological Pathway.**
(JPG)

**S19 Fig. Vitamin D-sensitive calcium signaling in depression (a) Biological Process, (b) Wiki Pathways Enrichment and (c) Reactome Pathways Enrichment.**
(JPG)

**S1 Table. Pharmacology and Phytochemistry of Plants used by Traditional Healers of Sikkim.**
(DOCX)

**S2 Table. List of active phytochemical constituents identified from thirty-two traditional bone-mending plants; 1,043 compounds were retrieved, 527 non-redundant molecules were retained, and PubChem SMILES were used for target prediction analyses.**
(XLSX)

**S3 Table. Summary of target genes obtained from database searches: 15,949 predicted targets, 7,612 bone-related disease genes, and 707 overlapping genes used for network construction, including protein–protein interactions, KEGG pathway mapping, and functional enrichment analyses.**
(XLSX)

## Acknowledgments

The Authors are thankful to the Department of Forest, Environment and Wildlife Management, Government of Sikkim, Gangtok, for issuing research permit (F. NO.: 78/GOS/FandED/RandE/139 dated August 3, 2022 and F.No.: 78/GOS/FEWMD/RandE/2027 dated November 7, 2023) and cooperation of officials for the completion of this research work. Furthermore, the Author would also like to thank Mr. Jayanta Barman, Sikkim University and the Administration and Account Staff of BRIC-IBSD, Imphal, for their constant help.

## Author contributions

**Conceptualization:** Lokesh Deb.

**Data curation:** Mukunda Anuj Sharma, Bharat Gopalrao Somkuwar, Parvin A Barbhuiya, Bhumika Gurung, Madhusmita Mahapatra, Firdous Fatima, Teresa Ningthoujam, Ashika Bhattarai, Bishal Tiwari, Lokesh Deb.

**Formal analysis:** Mukunda Anuj Sharma, Bharat Gopalrao Somkuwar, Lokesh Deb.

**Investigation:** Mukunda Anuj Sharma, Bharat Gopalrao Somkuwar, Parvin A Barbhuiya, Bhumika Gurung, Madhusmita Mahapatra, Firdous Fatima, Teresa Ningthoujam, Ashika Bhattarai, Bikash Rai, Pravin Kumar, Bishal Tiwari, Sancha Kumar Subba, Zyankit Lepcha, Purna Maya Gurung, Nandalal Khadka, Ratan Bahadur Tamang, Balbir Khati, Shribhakta Chettri, Ran Bahadur Rai, Hem Lall Sharma, Yam Bahadur Rai, Norzang Lepcha, Theptuk Lepcha, Pritiman Singh Chettri, Monraj Limboo, Theweng Gyenthen, Tulshi Pradhan, Prem Gurung, Prem Prashad Dhakal.

**Methodology:** Mukunda Anuj Sharma, Bharat Gopalrao Somkuwar, Firdous Fatima, Lokesh Deb.

**Project administration:** Nanaocha Sharma.

**Resources:** Mukunda Anuj Sharma, Bharat Gopalrao Somkuwar, Bhumika Gurung, Madhusmita Mahapatra, Ashika Bhattarai, Bikash Rai, Pravin Kumar, Bishal Tiwari, Sancha Kumar Subba, Zyankit Lepcha, Purna Maya Gurung, Nandalal Khadka, Ratan Bahadur Tamang, Balbir Khati, Shribhakta Chettri, Ran Bahadur Rai, Hem Lall Sharma, Yam Bahadur Rai, Norzang Lepcha, Theptuk Lepcha, Pritiman Singh Chettri, Monraj Limboo, Theweng Gyenthen, Tulshi Pradhan, Prem Gurung, Prem Prashad Dhakal.

**Software:** Bharat Gopalrao Somkuwar, Firdous Fatima.

**Supervision:** Nanaocha Sharma, Lokesh Deb.

**Validation:** Mukunda Anuj Sharma, Bharat Gopalrao Somkuwar, Lokesh Deb.

**Visualization:** Mukunda Anuj Sharma, Bharat Gopalrao Somkuwar.

**Writing – original draft:** Lokesh Deb.

**Writing – review & editing:** Mukunda Anuj Sharma, Bharat Gopalrao Somkuwar, Parvin A Barbhuiya, Teresa Ningthoujam, Ashika Bhattarai, Bishal Tiwari, Nanaocha Sharma.

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
