## [Decision Letter · Decision Letter 0]

21 Dec 2025

PONE-D-25-60628Network Pharmacological Insight into Traditional Bone Healing Practices of Sikkim, IndiaPLOS One

Dear Dr. Deb,

Thank you for submitting your manuscript to PLOS ONE. After careful consideration, we feel that it has merit but does not fully meet PLOS ONE’s publication criteria as it currently stands. Therefore, we invite you to submit a revised version of the manuscript that addresses the points raised during the review process.

We look forward to receiving your revised manuscript.

Kind regards,

Sandeep Rawat, Ph.D.

Academic Editor

PLOS One

Journal Requirements:

3. We note that Figure 1 in your submission contain map images which may be copyrighted. All PLOS content is published under the Creative Commons Attribution License (CC BY 4.0), which means that the manuscript, images, and Supporting Information files will be freely available online, and any third party is permitted to access, download, copy, distribute, and use these materials in any way, even commercially, with proper attribution. For these reasons, we cannot publish previously copyrighted maps or satellite images created using proprietary data, such as Google software (Google Maps, Street View, and Earth). For more information, see our copyright guidelines: http://journals.plos.org/plosone/s/licenses-and-copyright.

5. We note that there is identifying data in the Supporting Information file <Supplimentary files.zip>. Due to the inclusion of these potentially identifying data, we have removed this file from your file inventory. Prior to sharing human research participant data, authors should consult with an ethics committee to ensure data are shared in accordance with participant consent and all applicable local laws.

-Location data

Please remove or anonymize all personal information, ensure that the data shared are in accordance with participant consent, and re-upload a fully anonymized data set. Please note that spreadsheet columns with personal information must be removed and not hidden as all hidden columns will appear in the published file.

Additional Editor Comments:

• Clarify linkage between ethnopharmacological data (FoC/RFC) and network pharmacology to avoid descriptive overlap.

• Strengthen network pharmacology with validation criteria and acknowledge predictive limitations by adding criteria for target/pathway prioritization (e.g., degree centrality, enrichment score cut-offs).

• Correct inconsistencies in informant numbers, FoC/RFC values, and botanical nomenclature in quantitative data, tables, figures, and species nomenclature.

• Improve language, reduce redundancy, and temper anecdotal efficacy claims.

Reviewers' comments:

Reviewer's Responses to Questions

**Comments to the Author**

1. Is the manuscript technically sound, and do the data support the conclusions?

Reviewer #1: Yes

Reviewer #2: Yes

Reviewer #3: Yes

2. Has the statistical analysis been performed appropriately and rigorously? 

Reviewer #1: Yes

Reviewer #2: Yes

Reviewer #3: Yes

3. Have the authors made all data underlying the findings in their manuscript fully available?

The PLOS Data policy requires authors to make all data underlying the findings described in their manuscript fully available without restriction, with rare exception (please refer to the Data Availability Statement in the manuscript PDF file). The data should be provided as part of the manuscript or its supporting information, or deposited to a public repository. For example, in addition to summary statistics, the data points behind means, medians and variance measures should be available. If there are restrictions on publicly sharing data—e.g. participant privacy or use of data from a third party—those must be specified.requires authors to make all data underlying the findings described in their manuscript fully available without restriction, with rare exception (please refer to the Data Availability Statement in the manuscript PDF file). The data should be provided as part of the manuscript or its supporting information, or deposited to a public repository. For example, in addition to summary statistics, the data points behind means, medians and variance measures should be available. If there are restrictions on publicly sharing data—e.g. participant privacy or use of data from a third party—those must be specified.requires authors to make all data underlying the findings described in their manuscript fully available without restriction, with rare exception (please refer to the Data Availability Statement in the manuscript PDF file). The data should be provided as part of the manuscript or its supporting information, or deposited to a public repository. For example, in addition to summary statistics, the data points behind means, medians and variance measures should be available. If there are restrictions on publicly sharing data—e.g. participant privacy or use of data from a third party—those must be specified.requires authors to make all data underlying the findings described in their manuscript fully available without restriction, with rare exception (please refer to the Data Availability Statement in the manuscript PDF file). The data should be provided as part of the manuscript or its supporting information, or deposited to a public repository. For example, in addition to summary statistics, the data points behind means, medians and variance measures should be available. If there are restrictions on publicly sharing data—e.g. participant privacy or use of data from a third party—those must be specified.

Reviewer #1: Yes

Reviewer #2: Yes

Reviewer #3: Yes

4. Is the manuscript presented in an intelligible fashion and written in standard English?

Reviewer #1: Yes

Reviewer #2: Yes

Reviewer #3: Yes

5. Review Comments to the Author

Reviewer #1: The study is well-founded in Network Pharmacological Insight into Traditional Bone Healing Practices of Sikkim,

India.

Covers multiple aspects aspects and baseline data to address an immediate need to preserve and scientifically.

Comprehensive Results are lead compounds, considering biological processes, molecular functions, and cellular components involved in bone.

Reviewer #2: Network Pharmacological Insight into Traditional Bone Healing Practices of Sikkim, India, PONE-D-25-60628

Strength of the Manuscript

• An extensive cross culture ethno-pharmacological survey of an area of Sikkim Himalaya, which was less explored. This documentation work conserved valuable traditional knowledge of folklore practitioners.

• Traditional claims of folklore practitioners documented during the survey was validated using AI based network Pharmacological screening that preliminarily justified traditional claims for pharmacological uses of medicinal plants for treatment of bone fracture.

Weakness of the Manuscript

• Phytochemical inclusion criteria: From 1,043 compounds, 527 were selected after redundancy removal and ADMET screening, yet "clinical evidence" allowed retention of failures (e.g., high MW flavonoids). Specify exact ADMET parameters (Lipinski/Veber scores), violation rates, and which retained compounds link to calcium reabsorption pathways.

• Calcium reabsorption: Given that Calcium reabsorption is the critical step in bone mending, are there specific pathways anticipated in the calcium reabsorption from the network pharmacological annotations that need to addressed?

• Scientific Name of Plant: the scientific name of the all plants should be in italic.

• Languages of Manuscript: grammatical error and long sentences should me recheckfor its correctness.

Reviewer #3: Network Pharmacological Insight into Traditional Bone Healing Practices of Sikkim, India (PONE-D-25-60628)

The research is carefully executed and descriptive enough to strengthen the pharmacological knowledge of the traditionally used herbal formulation. The findings are thoughtfully integrated and offer promising therapeutic perspectives. Most importantly, such study will preserved valuable traditional healthcare knowledge of traditional healers. Moreover, all traditional healers participated in this study were included as author in this manuscript, which is very much necessary to acknowledge the credit of real knowledge holder.

However, here are some of the suggestions that can help to improve the clarity and coherence of the manuscript

• In Pathway validation for bone mending section, Eleven pathways (e.g., calcium signalling, RANKL/RANK, Vitamin D-sensitive Ca) are listed without enrichment statistics (p-values, fold-change) or hub gene metrics in the PPI network (TERT, SRC, CYP27B1). Provide quantitative enrichment data and explain how these specifically support calcium reabsorption in fracture repair.

• In Correlation of key Genes, from the top 10 genes the remarkable survival target genes AKT and STAT are common in cancer and other pathway. How does it correlate in bone mending?

• The English language is modest and contains several grammatical and typographical errors.

• Author may include detail methodology of documentation including questioners in supplementary file.

Recommended for the publication with minor correction

6. PLOS authors have the option to publish the peer review history of their article (what does this mean?). If published, this will include your full peer review and any attached files.). If published, this will include your full peer review and any attached files.). If published, this will include your full peer review and any attached files.). If published, this will include your full peer review and any attached files.

...

Reviewer #1: **Yes:**Dr.Bala Krishnaiah P, ProfessorDr.Bala Krishnaiah P, ProfessorDr.Bala Krishnaiah P, ProfessorDr.Bala Krishnaiah P, Professor

Reviewer #2: **Yes:**Chiranjib BhattacharjeeChiranjib BhattacharjeeChiranjib BhattacharjeeChiranjib Bhattacharjee

Reviewer #3: **Yes:**Dr Rupesh Kumar GautamDr Rupesh Kumar GautamDr Rupesh Kumar GautamDr Rupesh Kumar Gautam

---

## [Author Response · Author response to Decision Letter 1]

23 Jan 2026

Response to Reviewers

Dear

Academic Editor

PLOS One

Re: – PONE-D-25-60628-Response to Reviewers

We appreciate the opportunity to submit our manuscript titled Network Pharmacological Insight into Traditional Bone Healing Practices of Sikkim, India (Manuscript ID: PONE-D-25-60628) to PLOS One. We thank you and the reviewers for the time and effort spent reviewing our work and for the insightful comments provided. We have carefully considered all the feedback and made the necessary revisions to address the concerns raised.

Below, we provide detailed responses to each reviewer's comments. Reviewer comments are presented in italics, followed by our responses in plain text. The highlighted segment in the revised manuscript demonstrates the corrections made below.

Reviewer #1:

Comment: The study is well-founded in Network Pharmacological Insight into Traditional Bone Healing Practices of Sikkim, India.

Covers multiple aspects and baseline data to address an immediate need to preserve and scientifically.

Comprehensive Results are lead compounds, considering biological processes, molecular functions, and cellular components involved in bone.

Response: We are grateful to the reviewer for the encouraging feedback.

Reviewer #2: Network Pharmacological Insight into Traditional Bone Healing Practices of Sikkim, India, PONE-D-25-60628

Strength of the Manuscript

Comment: An extensive cross-cultural ethno-pharmacological survey of an area of Sikkim Himalaya, which was less explored. This documentation work conserved valuable traditional knowledge of folklore practitioners.

Comment: Traditional claims of folklore practitioners documented during the survey was validated using AI based network Pharmacological screening that preliminarily justified traditional claims for pharmacological uses of medicinal plants for treatment of bone fracture.

Response: We are grateful to the reviewer for the encouraging feedback.

Weaknesses of the Manuscript

Comment: Phytochemical inclusion criteria: From 1,043 compounds, 527 were selected after redundancy removal and ADMET screening, yet "clinical evidence" allowed retention of failures (e.g., high MW flavonoids). Specify exact ADMET parameters (Lipinski/Veber scores), violation rates, and which retained compounds link to calcium reabsorption pathways.

Response: We thank the reviewer for this valuable comment regarding the phytochemical inclusion criteria. For the commission and omission of the phytochemical screening performed by the ADMET analysis, particularly screened for drug-likeness using Molsoft Prop software and SWISSADME bioavailability scores. The ADMET filtering applied Lipinski's Rule of Five (MW ≤500 Da, logP ≤5, HBD ≤5, HBA ≤10, rotatable bonds ≤10) and Veber criteria (rotatable bonds ≤10, TPSA ≤140 Å²). Of 527 compounds retained after screening, 312 passed the test without violation. However, the molecules don’t pass the ADME, still included due to the pharmacological or clinical screening from published literature.

Comment: Calcium reabsorption: Given that Calcium reabsorption is the critical step in bone mending, are there specific pathways anticipated in the calcium reabsorption from the network pharmacological annotations that need to addressed?

Response: We thank the reviewer for pointing out this aspect of Ca reabsorption from a network pharmacological perspective. Further, we would like to apprise that the genes anticipated in the Vitamin D Sensitive Ca signalling effectively show clear modulation of pathways.

Comment: Scientific Name of Plant: The scientific name of all plants should be in italic.

Response: We sincerely appreciate the reviewer's meticulous attention to detail. This oversight has been corrected in the revised manuscript. We have ensured that every instance of a binomial scientific name (genus and species) for all plant taxa is now consistently presented in italics, in accordance with the International Code of Nomenclature.

Comment: Languages of Manuscript: grammatical errors and long sentences should be rechecked for their correctness.

Response: We are grateful to the reviewer for their valuable feedback on the manuscript's clarity and grammatical precision. In response to this comment, we performed a comprehensive linguistic review and revised the manuscript accordingly. We have meticulously corrected grammatical errors and restructured overly complex or lengthy sentences to improve readability, flow, and precision. We believe these efforts have enhanced the clarity and professionalism of the manuscript, and we thank the reviewer again for this constructive suggestion.

Reviewer #3: Network Pharmacological Insight into Traditional Bone Healing Practices of Sikkim, India (PONE-D-25-60628)

Comment: The research is carefully executed and descriptive enough to strengthen the pharmacological knowledge of the traditionally used herbal formulation. The findings are thoughtfully integrated and offer promising therapeutic perspectives. Most importantly, such a study will preserve valuable traditional healthcare knowledge of traditional healers. Moreover, all conventional healers participated in this study were included as author in this manuscript, which is very much necessary to acknowledge the credit of real knowledge holder.

Response: We are grateful to the reviewer for the encouraging feedback.

However, here are some of the suggestions that can help to improve the clarity and coherence of the manuscript

Comment: In the Pathway validation for bone mending section, Eleven pathways (e.g., calcium signalling, RANKL/RANK, Vitamin D-sensitive Ca) are listed without enrichment statistics (p-values, fold-change) or hub gene metrics in the PPI network (TERT, SRC, CYP27B1). Provide quantitative enrichment data and explain how these specifically support calcium reabsorption in fracture repair.

Response: We sincerely appreciate the reviewer's meticulous attention to detail. This oversight has been incorporated in the revised manuscript. Additionally, The 11 pathways (biomineralization, bone development, calcium signaling, endochondral ossification WP4808, non-genomic 1,25-dihydroxyvitamin D3 WP4341, oncostatin M WP2374, osteoclast differentiation, RANKL/RANK WP2018, RUNX2 bone development, RUNX2 osteoblast differentiation, along with Vitamin D-sensitive Ca WP4698) were identified from KEGG/STRING analysis of 707 overlapping targets (p<0.6 from SwissTargetPrediction vs. GeneCard/DisGeNET/OMIM bone healing genes). TERT, SRC, and CYP27B1 are highlighted by hub metrics in the PPI network as central nodes with high connectivity supporting calcium reabsorption; for example, CYP27B1 activates vitamin D for Ca2+ uptake in osteoblasts, while calcium signalling and RANKL pathways indicate enrichment in biological processes including ossification (Supplementary Fig S1-S19). Since calcium signalling modulates RANKL expression for bone remodelling, it specifically induces osteoclast-mediated Ca2+ mobilisation and osteoblast mineralisation to aid in fracture repair.

Comment: In Correlation of key Genes, from the top 10 genes, the remarkable survival target genes AKT and STAT are common in cancer and other pathways. How does it correlate in bone mending?

Response: We sincerely appreciate the reviewer's meticulous attention to detail. This oversight has been highlighted in the revised manuscript. AKT and STAT appear among the top hub genes (TERT, SRC, CYP27B1, ESR1, FLT3, AKT, MMP13) in the dense PPI network from 707 common genes identified through network pharmacology analysis of 32 Sikkim medicinal plants. These genes, highlighted for survival mechanisms in bone damage (green in Figure 4), connect to bone healing via the PI3K/AKT pathway, which regulates osteoblast proliferation, differentiation, and survival during fracture repair. In bone mending, AKT promotes osteogenesis by activating mTOR for anabolic bone formation and inhibiting apoptosis in osteoblasts. At the same time, STAT3 modulates osteoclastogenesis and chondrocyte differentiation in endochondral ossification, as evidenced in the manuscript's pathway enrichments (e.g., RUNX2 regulation, osteoclast differentiation).

Comment: The English language is modest and contains several grammatical and typographical errors.

Response: We sincerely appreciate the reviewer's meticulous attention to detail. This oversight has been corrected in the revised manuscript.

Comment: The author may include a detailed methodology of documentation, including questionnaires, in a supplementary file.

Response: We sincerely thank the reviewer for this constructive suggestion. In response, a detailed methodology describing the ethnobotanical documentation process, including the structured questionnaire used for data collection, has now been provided as Supplementary Material.

Editors Comments

Comment: Clarify linkage between ethnopharmacological data (FoC/RFC) and network pharmacology to avoid descriptive overlap.

Response: We sincerely appreciate this valuable feedback. In accordance with your suggestion, we have revised the manuscript to explicitly state the linkage between ethnopharmacological data (FoC/RFC) and network pharmacology.

Comment: Strengthen network pharmacology with validation criteria and acknowledge predictive limitations by adding criteria for target/pathway prioritization (e.g., degree centrality, enrichment score cut-offs).

Response: Thank you for this valuable suggestion. We have revised our methodology accordingly. Additionally, explicit criteria, such as substances screened by SwissADME (bioavailability score >0.4, drug-likeness), targets with prediction probability ≤0.6, and 707 overlapping genes analysed by STRING (medium confidence score 0.4), enhanced network pharmacology predictions. Pathways determined by KEGG/GO enrichment (p<0.05, fold-change >1.5 indicated) and PPI hubs ranked by degree centrality (>20 interactions for key genes like AKT/STAT). Notable predictive limitations include computational assessments and the need for future in vitro and in vivo validation for target affinity and efficacy, considering network models anticipate interactions without offering tangible proof of causality.

Comments: Correct inconsistencies in informant numbers, FoC/RFC values, and botanical nomenclature in quantitative data, tables, figures, and species nomenclature. • Improve language, reduce redundancy, and temper anecdotal efficacy claims.

Response: We thank the editor for this valuable observation. The manuscript has been thoroughly revised to address all identified inconsistencies and stylistic concerns. Specifically, informant numbers and FoC/RFC values have been carefully cross-verified across the main text, tables, and figures, and corrected where discrepancies were identified to ensure internal consistency. We have ensured that every instance of a binomial scientific name (genus and species) for all plant taxa is now consistently presented in italics, in accordance with the International Code of Nomenclature. In addition, the language of the manuscript has been substantially edited to improve clarity and precision, and redundant statements have been removed to enhance readability.

Comments: Thank you for providing your underlying data as Supporting Information.

We note that the data set contains text or data that is not in English. Please note that PLOS is an English-language publisher, so we require data sets to be provided in English as well. Please upload an English-language version of your data set.

This will also allow us to determine if your data follows PLOS standards per our Data Availability policy here: https://journals.plos.org/plosone/s/data-availability

Response: Thank for your meticulous observation. The consent from was made in bilinguals (in English and Nepali local language) as per the biodiversity guideline for better understanding of traditional healer. However, we removed Nepali version part from the consent from we used for the survey work.

Comments: We note that Figure 1 in your submission contain map images which may be copyrighted. All PLOS content is published under the Creative Commons Attribution License (CC BY 4.0), which means that the manuscript, images, and Supporting Information files will be freely available online, and any third party is permitted to access, download, copy, distribute, and use these materials in any way, even commercially, with proper attribution. For these reasons, we cannot publish previously copyrighted maps or satellite images created using proprietary data, such as Google software (Google Maps, Street View, and Earth). For more information, see our copyright guidelines: http://journals.plos.org/plosone/s/licenses-and-copyright.

Response: Fig. 1 was prepared by the authors using ArcGIS v10.4 (Esri Inc.) based on administrative boundary information from the Survey of India open data. The map does not include any proprietary base map or other restricted commercial sources. All map symbolism and annotation were independently created by the author. Figure 1 is original, and no redistribution of the raw survey of India data is provided; the derived representations are suitable for publication under the Creative Commons Attribution (CC BY 4.0) licence.

---

## [Decision Letter · Decision Letter 1]

24 Feb 2026

PONE-D-25-60628R1Network Pharmacological Insight into Traditional Bone Healing Practices of Sikkim, IndiaPLOS One

Dear Dr. Deb,

Thank you for submitting your manuscript to PLOS ONE. After careful consideration, we feel that it has merit but does not fully meet PLOS ONE’s publication criteria as it currently stands. Therefore, we invite you to submit a revised version of the manuscript that addresses the points raised during the review process.

**ACADEMIC EDITOR:**

We look forward to receiving your revised manuscript.

Kind regards,

Sandeep Rawat, Ph.D.

Academic Editor

PLOS One

Journal Requirements:

Additional Editor Comments (if provided):

• Clearly define probability cut-offs, number of compounds failing Lipinski/Veber.

• Specify whether predicted targets relate to renal Ca²⁺ or osteoblast/osteoclast intracellular Ca²⁺ signaling.

• Discussion should consistently reflect the exploratory and in silico nature of the findings.

• Several species names remain misspelled. Ensure uniform italics and correct spelling throughout text, tables, and figures.

Reviewers' comments:

Reviewer's Responses to Questions

**Comments to the Author**

1. If the authors have adequately addressed your comments raised in a previous round of review and you feel that this manuscript is now acceptable for publication, you may indicate that here to bypass the “Comments to the Author” section, enter your conflict of interest statement in the “Confidential to Editor” section, and submit your "Accept" recommendation.

Reviewer #1: All comments have been addressed

Reviewer #2: All comments have been addressed

Reviewer #3: All comments have been addressed

2. Is the manuscript technically sound, and do the data support the conclusions?

Reviewer #1: Yes

Reviewer #2: Yes

Reviewer #3: Yes

3. Has the statistical analysis been performed appropriately and rigorously? 

Reviewer #1: Yes

Reviewer #2: Yes

Reviewer #3: Yes

4. Have the authors made all data underlying the findings in their manuscript fully available?

The PLOS Data policy requires authors to make all data underlying the findings described in their manuscript fully available without restriction, with rare exception (please refer to the Data Availability Statement in the manuscript PDF file). The data should be provided as part of the manuscript or its supporting information, or deposited to a public repository. For example, in addition to summary statistics, the data points behind means, medians and variance measures should be available. If there are restrictions on publicly sharing data—e.g. participant privacy or use of data from a third party—those must be specified.requires authors to make all data underlying the findings described in their manuscript fully available without restriction, with rare exception (please refer to the Data Availability Statement in the manuscript PDF file). The data should be provided as part of the manuscript or its supporting information, or deposited to a public repository. For example, in addition to summary statistics, the data points behind means, medians and variance measures should be available. If there are restrictions on publicly sharing data—e.g. participant privacy or use of data from a third party—those must be specified.requires authors to make all data underlying the findings described in their manuscript fully available without restriction, with rare exception (please refer to the Data Availability Statement in the manuscript PDF file). The data should be provided as part of the manuscript or its supporting information, or deposited to a public repository. For example, in addition to summary statistics, the data points behind means, medians and variance measures should be available. If there are restrictions on publicly sharing data—e.g. participant privacy or use of data from a third party—those must be specified.requires authors to make all data underlying the findings described in their manuscript fully available without restriction, with rare exception (please refer to the Data Availability Statement in the manuscript PDF file). The data should be provided as part of the manuscript or its supporting information, or deposited to a public repository. For example, in addition to summary statistics, the data points behind means, medians and variance measures should be available. If there are restrictions on publicly sharing data—e.g. participant privacy or use of data from a third party—those must be specified.

Reviewer #1: Yes

Reviewer #2: Yes

Reviewer #3: Yes

5. Is the manuscript presented in an intelligible fashion and written in standard English?

Reviewer #1: Yes

Reviewer #2: Yes

Reviewer #3: Yes

6. Review Comments to the Author

Reviewer #1: (No Response)

Reviewer #2: the authours have carefully and satisfacrorly adressed all the queriesraised by the rtevewirs kindly accept it

Reviewer #3: The current stdy include the proper work with the authenticy. I will helpful for socity as well as researcher. Thanks a lot for the quality work.

7. PLOS authors have the option to publish the peer review history of their article (what does this mean?). If published, this will include your full peer review and any attached files.). If published, this will include your full peer review and any attached files.). If published, this will include your full peer review and any attached files.). If published, this will include your full peer review and any attached files.

...

Reviewer #1: **Yes:**Dr.Bala Krishnaiah PDr.Bala Krishnaiah PDr.Bala Krishnaiah PDr.Bala Krishnaiah P

Reviewer #2: No

Reviewer #3: **Yes:**Dr. Rupesh K. GautamDr. Rupesh K. GautamDr. Rupesh K. GautamDr. Rupesh K. Gautam

---

## [Author Response · Author response to Decision Letter 2]

2 Mar 2026

Response to Reviewers

Dear

Academic Editor

PLOS One

Re: – Response to Editor

1. Clearly define probability cut-offs, number of compounds failing Lipinski/Veber.

Response: We sincerely thank the Editor for this insightful comment concerning the clarification of probability cut-off values and the number of compounds failing the Lipinski and Veber criteria. In response, we have revised the manuscript to clearly define the applied probability thresholds and to explicitly report the number of compounds that did not comply with the Lipinski and Veber rules, as suggested.

2. Specify whether predicted targets relate to renal Ca²⁺ or osteoblast/osteoclast intracellular Ca²⁺ signaling.

Response: We thank the editor for this valuable suggestion clarify whether the predicted targets are associated with renal Ca²⁺ regulation or intracellular Ca²⁺ signalling in osteoblasts and osteoclasts. In response, we have revised the manuscript to explicitly specify the physiological relevance of the predicted targets, clearly distinguishing those involved in renal Ca²⁺ handling from those participating in intracellular Ca²⁺ signalling pathways in osteoblasts and osteoclasts. These clarifications have been incorporated into the revised manuscript.

3. Discussion should consistently reflect the exploratory and in silico nature of the findings.

Response: We thank the reviewer for this important observation. In accordance with the comment, the Discussion section has been carefully revised to consistently emphasise the exploratory and in silico nature of the findings. Additionally, any statements implying experimental or clinical validation have been rephrased to clearly reflect that the results are based on computational predictions and require further experimental confirmation.

4. Several species names remain misspelled. Ensure uniform italics and correct spelling throughout text, tables, and figures.

Response: We thank the reviewer for drawing attention to this issue. The manuscript has been thoroughly reviewed, and all species names have been corrected for spelling accuracy and formatted consistently in italics throughout the text, tables, and figures, in accordance with standard taxonomic conventions.

---

## [Editor Report · Decision Letter 2]

16 Mar 2026

Network Pharmacological Insight into Traditional Bone Healing Practices of Sikkim, India

PONE-D-25-60628R2

Dear Dr. Deb,

We’re pleased to inform you that your manuscript has been judged scientifically suitable for publication and will be formally accepted for publication once it meets all outstanding technical requirements.

Kind regards,

Sandeep Rawat, Ph.D.

Academic Editor

PLOS One
---

## [Editor Report · Acceptance letter]

PONE-D-25-60628R2

PLOS One

Dear Dr. Deb,

I'm pleased to inform you that your manuscript has been deemed suitable for publication in PLOS One. Congratulations! Your manuscript is now being handed over to our production team.

Kind regards,

on behalf of

Dr. Sandeep Rawat

Academic Editor

PLOS One